# Low Alanine-Aminotransferase Blood Activity Is Associated with Increased Mortality in Chronic Lymphocytic Leukemia Patients: A Retrospective Cohort Study of 716 Patients

**DOI:** 10.3390/cancers15184606

**Published:** 2023-09-17

**Authors:** Tamer Hellou, Guy Dumanis, Arwa Badarna, Gad Segal

**Affiliations:** 1Hematology Division, Sheba Medical Center, Faculty of Medicine, Tel-Aviv University, Ramat Gan 5266202, Israel; tamer.hellou@sheba.health.gov.il; 2Adelson School of Medicine, Ariel University, Ariel 4070000, Israel; 3Sheba Medical Center, School of Medicine, Karazin Kharkiv National University, 61022 Kharkiv, Ukraine; arwa.bdarneh@gmail.com; 4Education Authority, Sheba Medical Center, Faculty of Medicine, Tel-Aviv University, Ramat Gan 5266202, Israel

**Keywords:** CLL, ALT, sarcopenia, survival, frailty, prognosis

## Abstract

**Simple Summary:**

For many patients with chronic diseases, both the prognosis (future prospects) and the optional treatments are dependent not only on the nature of disease but also on the patients themselves: how robust or frail they were prior to their illness. The degree of frailty (opposite of robustness) can be measured by a simple blood test (called ALT). In this research, we found that low ALT, suggestive of frailty, is associated with worse outcomes among CLL patients. We encourage physicians, therefore, to incorporate this blood test in their appreciation of CLL patients.

**Abstract:**

Background: Chronic lymphocytic leukemia (CLL) is one of the most common hematologic malignancies, especially among elderlies. Several prognostic scores are available that utilize the characteristics of patients’ blood counts and cytogenetic anomalies—all are features of the disease rather than of the patient. Addressing the route of personalized rather than precise medicine, we refer to the assessment of patients’ status of sarcopenia and frailty. Low alanine aminotransferase (ALT) was already shown to function as a surrogate marker for sarcopenia and frailty. We decided to find a possible correlation between low ALT values and poor prognosis of CLL patients. Patients and Methods: This is a retrospective cohort study of CLL patients treated in a large, tertiary medical center, as outpatients or inpatients. Their frailty status was evaluated in a retrospective manner. We defined patients with ALT below 12 IU/L as frail and divided our cohort into two groups including a low ALT level group (ALT < 12) and a normal ALT level group (ALT ≥ 12). Results: Overall, our final analysis included 716 CLL patients, of which 161 (22.5%) had ALT levels lower than 12 IU/L. There was no significant difference in patients’ age between the two groups. Patients with the low ALT had a lower hemoglobin concentration (median 10.8 g/dL [IQR = 2.7] vs. 12.2 [IQR = 3.1]; *p* < 0.001) and a higher proportion of patients were classified as Binet C score [48.4% vs. 31.1%]; *p* < 0.001). Frail CLL patients’ survival was significantly shorter when compared to non-frail patients, in both the univariate [HR = 1.6 [95% confidence interval, CI 1.23, 2.0]; *p* < 0.01] and multivariate analyses [HR = 1.3 [95% CI 1.0, 1.7]; *p* = 0.03]. Conclusions: Sarcopenia and frailty assessment, based on blood ALT measurements, could potentially point out differences in CLL patients’ prognoses. Such assessment could serve the purpose of treatment personalization of CLL patients.

## 1. Background

### 1.1. Chronic Lymphocytic Leukemia as a Common Hematologic Malignancy

CLL is characterized by clonal B cell proliferation equal to or higher than five thousand B-lymphocytes per microliter in the peripheral blood, for the duration of at least three months [1]. CLL is the most common hematologic malignancy in the elderly population with a median age at diagnosis of 72 years [2]. CLL, frequently diagnosed and initially managed as an indolent malignancy, negatively affects CLL patients’ overall survival [3]. As stated earlier, asymptomatic patients are commonly observed without intervention while symptomatic patients, fulfilling laboratory and clinical criteria (relating to disease characteristics thereby referred as indices of precise medicine), are candidates for an expanding array of advanced therapeutics [4]. For more than forty years the backbone systematic prognostic assessment of CLL patients was the Binet and Rai staging systems, emphasizing mainly the clinical heterogeneity course of the disease [5,6]. Recent years have brought a deeper understanding of the biologic and molecular aberrations contributing to the pathogenesis of CLL, leading to the development of new prognostic tools [7]. Currently, the International Prognostic Index for patients with chronic lymphocytic leukemia (CLL-IPI) is used, mainly for predicting the time to first treatment [8]. Treatment decisions are mainly based on the International Workshop on chronic lymphocytic leukemia (iwCLL) guidelines, published in 2018 [9]. 

The classification tools mentioned above are used for prognostication and treatment decisions, dealing mainly with disease characteristics—in other words, these are tools for executing the concepts of precise medicine. Such precision assessment tools lack aspects relating to the fitness of patients, potentially classifying them as robust on one hand or frail on the other hand, which would endow clinicians with aspects of personalized, rather than only precise, medicine. Currently, such personalization relies mainly on clinical judgment based on comorbidities assessment tools and their impact on our patients, like the cumulative illness rating scale (CIRS) [10] and the eastern cooperative oncology group performance status (ECOG PS) [11]. Most such tools are based on “eyeballing” the patients and subjectively assessing their potential stamina/capability of withstanding aggressive therapeutic measures. 

### 1.2. Sarcopenia and Frailty in CLL Patients

Hematologic malignancies in general, and CLL in particular, pose a challenge to clinicians treating the aged population, relating to their chronological age and simultaneously trying to relate to their biological age. Sarcopenia and frailty assessment in these patients could help clinicians make better treatment decisions relating to the prognosis of such patients [12]. Frailty is defined as a decline in physiologic reserve independent of the chronological age, making frail patients more vulnerable to any stressor, especially haemato-oncologic diseases and their treatments [13]. There is a worldwide agreement on the importance of frailty assessment especially in the field of oncology, but there are difficulties in identifying these patients in the daily clinical care, due to the heterogeneity of the phenotypic manifestation and the multiple existing definitions [14]. Sarcopenia is characterized by quantitative and qualitative muscle decline, in both mass and function respectively, which lead to poor physical function/capacity [15]. Muscle atrophy and physical deficits are shared characteristics in both the frailty and sarcopenia models, which led Marzetti et al. to merge these two conditions into a new clinical entity called physical frailty and sarcopenia [16,17]. 

Frailty and sarcopenia assessment are the backbone of treatment personalization in particular and the concept of personalized medicine in general. Numerous studies and publications fortified the logical basis and the factual basis of sarcopenia and frailty assessment in many chronic conditions. Nevertheless, there are not many available, easy-to-use clinical tests and parameters for the purpose of sarcopenia and frailty assessment. 

### 1.3. Low Alanine Aminotransferase as a Biomarker for Syndromes of Sarcopenia and Frailty

Alanine amino transferase, also known as SGPT—serum glutamic pyruvic transaminase—is an enzyme responsible for the transamination reaction of alanine and 2-oxoglutarate, generating both pyruvate and glutamic acid. This fundamental enzyme therefore plays a key role in the essential metabolism of glucose and amino acids. ALT catalyzes a bidirectional molecular process in the skeletal muscles and liver. Since ALT activity in the liver is 3000 times higher than in the blood, its main purpose in clinical medicine is to rule out hepatocellular injury. The activity of ALT in tissues other than the liver, like skeletal muscles, is much lower. The catalytic activity of ALT is facilitated by pyridoxal 5 phosphate, a metabolic derivative of vitamin B6, acting as a co-factor for ALT. Therefore, pyridoxal 5 phosphate is added to the serum test tubes in order to maximize the catalytic activity of ALT in the tube. 

In the past 10 years, many publications have been related to the assessment of sarcopenia and frailty by using ALT blood activity measurements as a biomarker. It has been shown that in patients without evidence of hepatitis, i.e., with ALT within the normal range, low-normal levels of ALT activity in the peripheral blood are associated with decreased mass of skeletal muscle when correlated to gender and age, and is also associated with decreased survival in the general population and associated with negative clinical outcomes in many different patient populations: patients with congestive heart failure for whom ALT values lower than 10 IU/L had a 1.22 hazard ratio for mortality during the follow-up period (CI = 1.1–1.4; *p* < 0.001), and patients with ischemic heart disease. In a multivariate regression model, ALT values equal to or lower than 10 IU/L were associated with increased mortality (HR 2.1, 95% CI 1.5–3), hospitalized COVID-19 patients, and patients suffering from chronic obstructive pulmonary disease. The crude hazard ratio for mortality in this population of patients with ALT levels lower than 11 IU was 2.4 (95% CI; 1.6–3.5). The increased risk of mortality remained significant after adjustment for age, weight, creatinine, albumin concentration and cardiovascular diseases (HR = 1.8; 95% CI 1.1–3.1, *p* < 0.05), patients undergoing rehabilitation after surgical repair of hip fractures (for whom a Cox regression analysis showed that low ALT blood levels prior to rehabilitation to be associated with increased one-year mortality (hazard ratio 1.9, 95% CI 1.08–3.3)), patients suffering from solid malignancy such as lung cancer, and also patients suffering from myelodysplastic syndrome (the hazard ratio for mortality in cases of ALT values lower than 12 IU/L was 1.2, 95% CI: 1.0–1.6, *p* = 0.04) [18,19,20,21,22,23,24,25,26]. Peltz-Sinvani et al. retrospectively analyzed the Bezafibrate Infarction Prevention study, regarding the association between low ALT and long-term survival. They included 6575 patients without known liver damage, followed for a median period of 22.8 years. The cumulative probability of all-cause mortality was significantly higher for those with low ALT when compared with those with higher ALT (65.6% vs. 58.4%; log-rank *p* < 0.001). A multivariate analysis, adjusted for multiple other well-known and accepted predictors of mortality, showed that low ALT was independently associated with an 11% higher long-term mortality, HR 1.11 [95% CI: 1.0–1.2; adjusted *p* < 0.01]. The authors concluded that low ALT levels are associated with increased long-term mortality in middle aged patients with stable coronary heart disease [27].

All of the above studies demonstrated a clear and statistically significant association between low-normal ALT values, in the range of 10 to 17 IU/L, and lower total straited muscle mass, sarcopenia, and markers for functional frailty as demonstrated by lower scores in validated questionnaires such as the FRAIL questionnaire. These lower values of ALT were also associated with lower chances of rehabilitation and shortened survival in heterogenous patient populations.

The aim of the current study was to assess the feasibility of diagnosing sarcopenia and frailty by using low ALT levels as a biomarker in a population of CLL patients. This ability will potentially allow hematology physicians to use future therapies with better treatment personalization of their CLL patients.

## 2. Patients and Methods

### 2.1. Patient Population

The patient population of the current study included patients diagnosed with CLL that were treated in a large, tertiary medical center, either as outpatients or inpatients. Following approval by the local review board, approval number SMC-23-0339, all patients’ characteristics were drawn from their electronic medical records, which are routinely used for clinical purposes and are, therefore, appropriate for withdrawing reliable data relating to their background and clinical characteristics. A total of 1458 CLL patient records were identified, of which 809 patients had a relevant record of their ALT value, i.e., at the time of diagnosis, not affected by later potential effects of therapies. In the above patients, we excluded those with ALT levels higher than 40 IU since these levels are predominantly a result of ALT originating from damaged liver cells and various types of acute and chronic hepatitis, rather than serving an indicator for their striated muscles’ mass. The final cohort includes 716 patients with ALT levels, within the normal limits, that were attained at the time of CLL diagnosis. Out of this cohort, the patients were classified into two main categories: above and equal to or below an ALT level of 12 IU/L, a normal ALT level group, and a low-normal ALT level group, respectively, referring to previous publications using this value as a cutoff for sarcopenia and frailty diagnosis. Figure 1 details the above patient consort flow and exclusion diagram.

### 2.2. Statistical Analysis

Baseline demographic and clinical data of patients were retrieved from the electronic medical record. We correlated the ALT values of patients alongside other clinical and laboratory characteristics with their survival as they appear in the national Israeli population registry. All variables were assessed for normality of distribution: in cases where there was a normal distribution, we described those parameters using means ± standard deviation (SD). Whenever the distribution of parameters was non-normal, they were described using medians with interquartile range (IQR). Categorical parameters were described using their frequency (%). Continuous data were compared with the student’s t-test when normally distributed, and with a non-parametric Wilcoxon signed-rank test when normality was rejected by a Shapiro–Wilk test. Chi-square tests or the Fisher exact test were applied whenever categorical data were compared. Crude survival was described by using a log-rank test, graphically described as a Kaplan–Meier curve. We used univariate Cox regression modelling in order to determine the unadjusted hazard ratio for the primary outcome. A multivariate model was constructed in order to examine both correlations and in order to control for possible confounders. An association was considered statistically significant for a two-sided *p* value lower than 0.05. All analyses were performed using R software version 4.2.0, R foundation for statistical computing.

## 3. Results

The final study population included 716 patients after the exclusion of patients without documented ALT and those with ALT levels higher than 40 IU/L, as illustrated in Figure 1. The final study cohort was divided into two categories according to ALT levels—lower than, or equal to and higher than 12 IU/L. We picked the ALT value of 12 IU on the basis of our groups’ previous experience: when investigating an older, potentially more frail population of patients, as it is with the CLL patients’ population, we use the lower values of ALT in order to define the group of frail patients. The low-normal ALT level group, with an ALT lower than 12 IU/L, had 161 patients (22.5%) with a median ALT level of 10 [IQR = 3 IU/L] compared to the normal ALT level group with a median ALT level of 18 [IQR =9 IU/L] (*p* < 0.001). The median age of the low-normal ALT level group was higher than the median age of the normal ALT level group (77.58 [IQR = 14.3] vs. 75.05 [IQR = 14.8] years), but this difference was not statistically significant. Patients in the low-normal ALT level group had a statistically significant lower weight (median 70 [IQR = 18.2] vs. 75 [IQR = 19] Kg; *p* = 0.014). Further, the low ALT level group had higher creatinine levels (median 1.0 [IQR = 0.45] vs. 0.97 [IQR = 0.4] mg/dL; *p* = 0.04); they also had lower hemoglobin concentrations (median 10.8 [IQR = 2.7] g/dL vs. 12.2 [IQR = 3.1] g/dL; *p* < 0.001); had lower platelet counts (median of 142 [IQR = 96] 10^9^/L vs. 164 [IQR = 101] 10^9^/L; *p* = 0.005); and were more likely to be categorized as Binet C score (48.4% vs. 31.1%; *p* < 0.001). There were no significant differences in the background diseases between the two groups as detailed in Table 1.

The whole cohort patients’ median survival time was 5.78 years (95% CI [4.9, 7.3]). Figure 2 shows the Kaplan–Meier curve for crude survival analysis according to ALT levels: the low-normal ALT level group had a statistically significant lower median overall survival time compared to the normal ALT level group (3.3 [IQR = 3.1] vs. 6.7 [IQR = 3.1] years; *p* < 0.001). Low ALT levels were associated with a statistically significant 56% increase in overall mortality (95% CI [1.2, 2.0]; *p* < 0.01), Table 2.

In our multivariate model (Table 2), we controlled for age, Binet C score, and creatinine concentration. We included those parameters in our multivariate model on the basis of common practice and results of our univariate analysis: the negative, inverse, correlation between ALT levels and mortality rates remains statistically significant [HR = 1.3, [1.0, 1.7]; *p* = 0.031] even after being incorporated into this multivariate model.

## 4. Discussion

The oncologic world in general, and the haemato-oncologic realm in particular, are experiencing advancements and a revolution in the treatment armamentarium, so we expect to see more patients in advanced age managed with relatively specialized therapeutics and enrolled into clinical trials in the near future [28]. For these purposes, we need solid assessment tools to make it easier for clinicians to identify those patients that are at relatively higher risk for complications relating to their age group, namely those who are frail [29].

Sarcopenia and frailty represent related syndromes, defined mainly in the realm of clinical geriatrics. The fact that someone has a low striated muscle mass relative to his age and gender group defines sarcopenia and his/her resultant lower functional capacity defines frailty. Both syndromes are associated with an increased risk of falls, hospitalizations, and death [16,17]. In contrast to the definition of “precise medicine”, which aims at the nature of the diseased tissue, sarcopenia and frailty relate to “personalized medicine” which aim at the gestalt of the patients’ body [26].

In the field of clinical haemato-oncology in general, and when dealing with CLL patients in particular, in an elderly patient population, physicians should concentrate on the means for the objective assessment of sarcopenia and frailty prior to taking significant therapeutic decisions. In the current study, we have shown that frail CLL patients—classified as those who have a low-normal levels of ALT activity in their blood—are subjected to shorter survival. The high availability of ALT measurements, as part of routine biochemical assessments, make this tool highly accessible for clinical use: ALT measurements are part of routine chemistry blood tests in almost every medical institution and are therefore available for clinicians in both retrospective analyses and prospective research. Assimilating such a simple biomarker into the routine assessment of CLL patients for sarcopenia and frailty could enable clinicians to make better decisions.

The availability of ALT measurement as a simple blood test makes it an easy-to-use tool for assessing a patient’s robustness/frailty, which can be an additional step in the general assessment of patients before treatment decisions. This is a potential step into assimilating a personalized approach to hematologic patients’ prognostication, alongside well-established parameters and scoring systems that relate to the disease itself, therefore considered elements of precision medicine. One such precision medicine tool is the CLL-IPI scoring system.

## 5. Conclusions

In our study population of CLL patients, which measured ALT blood levels of activity (in International Units), excluding those patients with higher-than-normal levels of ALT blood activity due to mainly hepatic problems, low ALT levels measured at the diagnosis of CLL patients were associated with poor prognosis and shortened survival. We ascertained this observation with both a crude survival analysis and after taking into consideration potential confounders appearing in both univariate and multivariate analyses. Low ALT enabled definitive and statistically significant separation of two CLL patient groups according to their prognosis.

Applying the concept of prognosis and treatment personalization using the ALT values of activity related to the pre-morbid condition of patients gives clinicians a valuable assessment of the robustness of their patients prior to taking significant treatment decisions.

## 6. Limitations

This study was conducted in a single center in the format of a retrospective study. Therefore, our results and conclusions should be further investigated in other patient populations before including our conclusions and recommendations into global guidelines of clinical practice. Future prospective studies, assessing therapeutic interventions for CLL patients, should be undertaken while guiding treatment decisions on the basis of both precise and personalized patient characteristics.

## Figures and Tables

**Figure 1 cancers-15-04606-f001:**
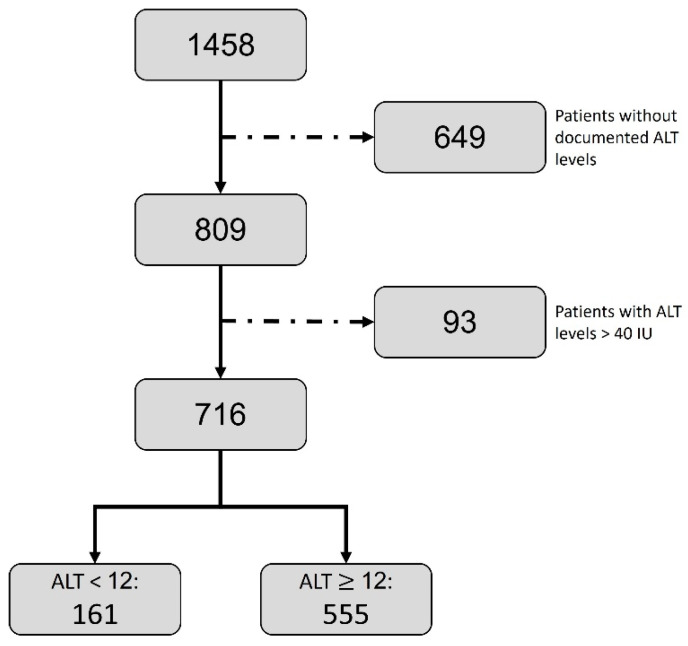
CONSORT flow of patients.

**Figure 2 cancers-15-04606-f002:**
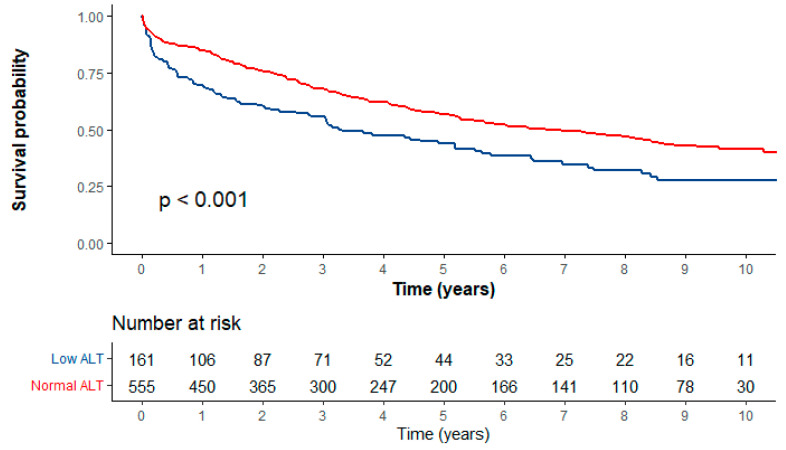
Kaplan–Meier survival analysis.

**Table 1 cancers-15-04606-t001:** Demographic and clinical features—all patients.

Whole Cohort N = 716	ALT < 12 IU/L N = 161 [22.5%]	ALT ≥ 12 IU/L N = 555 [77.5%]	*p* Value
ALT (IU/L); median [IQR]	10 [8–11]	18 [15–24]	<0.001
patient demographics			
Age (years); median [IQR]	77.6 [68.6–82.9]	75.05 [66.8–81.6]	0.05
Weight (Kg); median [IQR]	70 [61.5–80]	75 [64–83]	0.01
laboratory parameters			
Creatinine (mg/dL); median [IQR]	1.02 [0.8–1.3]	1.0 [0.8–1.2]	0.04
HB (g/dL), median [IQR]	10.8 [9.5–12.2]	12.2 [10.5–13.6]	<0.001
PLT (10^9^/L); median [IQR]	142 [97–193]	164 [114–215]	0.005
CLL characteristics			
Binet C; N (%)	78 (48.4)	170 (31.1)	<0.001
background disease			
CKD; N (%)	24 (14.9)	60 (10.8)	0.2
Solid malignancy; N (%)	41 (25.5)	124 (22.3)	0.47
Diabetes mellitus; N (%)	26 (16.1)	100 (18)	0.667
CHF; N (%)	19 (11.8)	69 (12.4)	0.937
IHD; N (%)	28 (17.4)	107 (19.3)	0.671
AF; N (%)	18 (11.2)	90 (16.2)	0.148
Stroke; N (%)	11 (6.8)	27 (4.9)	0.435
Dementia; N (%)	5 (3.1)	11 (2)	0.585
COPD; N (%)	11 (6.8)	36 (6.5)	1

ALT = alanine transaminase; HB = hemoglobin; PLT = platelets; CKD = chronic kidney disease; CHF = chronic heart failure; IHD = ischemic heart disease; AF = atrial fibrillation; COPD = chronic obstructive pulmonary disease.

**Table 2 cancers-15-04606-t002:** (**a**) Univariate analysis. (**b**) Multivariate analysis.

**(a)**		
**Patient Attribute**	**HR [95% CI]**	***p* Value**
ALT (IU/L) < 12	1.56 [1.23–1.97]	<0.001
**(b)**		
**Patient Attribute**	**HR [95% CI]**	***p* Value**
ALT (IU/L) < 12	1.31 [1.02–1.66]	0.031
Age (years)	1.01 [1.00–1.02]	0.005
Binet C	2.06 [1.66–2.54]	<0.001
Creatinine (mg/dL)	1.17 [1.04–1.33]	0.011

## Data Availability

No new data were created or analyzed in this study. Data sharing is not applicable to this article.

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
