# Peer review of "Low Alanine-Aminotransferase Blood Activity Is Associated with Increased Mortality in Chronic Lymphocytic Leukemia Patients: A Retrospective Cohort Study of 716 Patients"

_cancers, 2023, doi:10.3390/cancers15184606_

Round 1
Reviewer 1 Report
This large retrospective analysis investigated low ALT levels at diagnosis as an unfavorable prognostic marker in CLL patients. This reviewer wonders:
- almost half of all CLL pts were excluded due to missing ALT information; this is somewhat hard to believe and could harbor a potential bias. The authots should explain what was done to keep this level of missing values as low as possible.
- It remains unclear why pts with leveks > 40 have been excluded; again, a possible bias ?
- Finally, the cut-off of 12 remains arbitrary; was this the result of many rounds of analysis? Or a predefined cut-off?
- a number of parameters are significant as well (Hb; PLT; creatinine) which could be manifestations of more advanced disease; this would challenge the conclusions of "frailty", but rather suggest more advanced disease; the authors should conclude.
- Were genetic/ FISH parameters assessed ?
Author Response
Dear Reviewer,
On behalf of my team and myself I thank you very much for your efforts and comments that significantly improve our manuscript. Attached please find our point-by-point answers to your comments and the resultant changes. We hope that these will be satisfactory.
Best Regards,
Prof. Gad Segal, MD

Reviewer 2 Report
The authors reported that low blood ALT was associated with lower hemoglobin concentration, higher Binet C score, and shorter survival in patients with CLL, and concluded that low ALT may be a prognostic marker of CLL and useful to determine the time for initiating treatment of CLL. Although interesting, this manuscript has a number of major problems.
Major comments/
1. Blood ALT at the time of the CLL diagnosis appears one time measurement. Practically, the value of blood tests is rather fluctuating. Is it reasonable to determine low ALT by single measurement? Alternatively, how do the authors think reproducibility of the low ALT in individual patients?
2. The Background section is too long and insistent. Please simplify and shorten it by one-half.
3. Conclusion (page 7) is like discussion but not definite conclusion. Large part of this section should be moved to Discussion section. Please draw definite results in this section.
4. In Discussion section, please avoid duplication between issues in Background section, and concentrate to the significance of the results obtained in this study.
5. Figures 1 and 2, especially in Figure 2, lack figure legends. This is a major fault as a scientific paper.
Minor comments:
1. Abbreviations are largely confusing. The authors should keep “full-term followed by abbreviation in the parenthesis and use only abbreviation thereafter” throughout the manuscript. The abbreviation “TIFT” is not needed in page 2.
2. Please unify amongst/among. Furthermore, “in” is better than amongst throughout the manuscript.
3. Please avoid “patients’ survival/populations/etc.”, and try better English writing throughout the manuscript.
4. Page 3: Please don’t include a sentence in the parenthesis; only data/the number should be written in the parenthesis.
5. Please don’t use capital letter at the head of word unless the word is proper noun.
6. P lease write all the number of results in one decimal place such as 10.8 g/dL but not 10.77 g/dL throughout the text and Table 1 except for Table 2.
7. Tables 1 and 2: Please indicate IQR and 95%CI such as [8-11] but not [8, 11].
8. The unit of platelet count should be internationally indicated such as 142×109/L but not 142 K/mL.
9. Page 3, line 11: What is P-5-P? Is this abbreviation?
10. Page 2, line 5: “which many times is diagnosed” doesn’t make a sense.
11. Page 5: 12IU/L→12 IU/L in all cases.
Please refer to my comments, especially minor comments.
Author Response
Dear reviewer,
On behalf of my team and myself, I would like to thank you for your efforts and comments that significantly improve our manuscript. We hope that the corrections made will be satisfactory.
Best Regards,
Prof. Gad Segal, MD

Round 2
Reviewer 1 Report
The issues of this reviewer have been addressed.
Author Response
We thank you for your review.
Reviewer 2 Report
The authors revised the previous manuscript; however, the revision is not complete and to be improved for the publication in terms of basic problem of the manuscript preparation.
Major comments/
1. The legend for Figure 2 may be the paragraph from line 666 to 673. However, the last sentence of this paragraph is the legend for Table 2a but not for Figure 2. Please move this sentence for that of Table 2a.
1. Abbreviations are still confusing. Please correct throughout the manuscript. For example, ALT in Abstract and page 3, line 264, should be first written in full-term, then abbreviate it.
2. Page 3: Please don’t include a sentence in the parenthesis; only data/the number should be written in the parenthesis. This issue has not been improved.
3. Please don’t use capital letter at the head of word unless the word is proper noun. Also, this issue has not been fully improved.
4. Page 3, line 274: “P-5-P” may be an abbreviation of the term in the preceding sentence. If so, please write as “full-term then abbreviation in the parenthesis”.
5. Page 2, line 149: The sentence “CLL, which is diagnosed…” is not grammatically correct.
6. A sentence (?) in page 3, from line 289 to 297, “This increased risk…(18-26), does not make a sentence. Please correct making adequate sentences.
7. In page 5, a sentence from line 594 to 595, “In cases…(%)”, is strange. What is “(IQR..”?
Minor comments:
1. Page 1, line 74: 10.8 gr/dL→10.8 g/dL.
2. Page 3, line 286: 10 IU/l→10 IU/L.
3. Page 5, last sentence: table 1→Table 1.
Please refer major and minor comments.
Round 3
Reviewer 2 Report
The manuscript has been resonably improved at the second revision.
Still minor editing of English writing is required.